# Development and Validation of a Clinical Practicum Transition Shock Scale (CPT-Shock) for Korean Nursing Students

**DOI:** 10.3390/healthcare11202789

**Published:** 2023-10-21

**Authors:** Soo-Yeon Kim, Yeong-Ju Ko

**Affiliations:** 1Department of Nursing, Deagu Haany University, Hanuidae-ro 1, Gyeongsan-si 38610, Republic of Korea; sooyeonkim@dhu.ac.kr; 2Department of Nursing, Cheju Halla University, Halladaehak-ro, Jeju-si 63092, Republic of Korea

**Keywords:** clinical practicum, nursing student, transition shock

## Abstract

Clinical practice is an important educational tool for nursing students, and their emotions during clinical practice should be accurately understood. This study aimed to develop and verify the validity and reliability of a clinical practicum transition shock scale (CPT-shock) to measure Korean nursing students’ emotional responses when transitioning from a theoretical learning process to clinical practice. This research design was a methodological study and the content, construct, criterion validity, and reliability of the items were verified. Content validity was evaluated by seven experts. The preliminary survey was conducted on 24 nursing students, and the factor analysis was conducted on 331 nursing students in various regions of Korea. Confirmatory factor analysis confirmed the model fit (χ^2^/df = 1.741, GFI = 0.930, AGFI = 0.902, NFI = 0.869, TLI = 0.923, CFI = 0.938, RMR = 0.035, RMSEA = 0.047) and established discrimination and convergence validity. Positive and negative correlations were found with existing transition shock (r = 0.779) and clinical practice adaptation (r = −0.505), respectively (*p* < 0.001), thus establishing criterion validity. The reliability was good, with a Cronbach’s α of 0.85. The clinical practicum transition shock scale reflects nursing students’ practice environment and is expected to accurately measure nursing students’ unique emotional shock.

## 1. Introduction

Clinical practice experience is essential in nursing education, where learned knowledge is transformed from “knowledge” to “performance” [1]. Additionally, through practice, nursing students learn that nursing is not merely technical and that it also consists of developing nursing professionalism [2]. Although various teaching techniques such as high-fidelity simulations and virtual reality programs are being developed [3,4], clinical practice education is still considered important. As nursing is both a science and a practice based on understanding human beings, the real-world experience and patient interaction provided by clinical practice cannot be completely replaced [5].

Nursing students experience many changes during the transition from theoretical to practical education. “Transition shock” is an emotional response that may occur during the transition process and includes confusion, doubt, and loss of support [6]. The emotional shock is caused by physical changes in the learning environment from school to hospital, and the need to adapt to a nurse’s working hours and the associated lifestyle. Additionally, although students receive practical training in hospitals, patients or their guardians often expect them to act as caregivers. The experience of conflict between being a nurse and being a student in this ambiguous role is expected to contribute to the transition shock [7]. Moreover, unlike the educational institution environment where there is one professor for each class, in hospitals, students have to adapt to following different nurses on duty as shifts change [8].

The transition shock is very important for clinical entry as it is an adjustment period for individuals to adapt to change [9]. The transition shock period for nurses is estimated to last approximately two to four months [9]. However, in the case of nursing students, there is no significant difference in the transition shock between grades 3 and 4 (when clinical practice begins); therefore, it can be considered to last for about two years [10]. It is therefore necessary to precisely define the struggles that students experience in practice, determine their level of experience, and help them cope with the two-year adjustment period.

In previous studies, most of the emotions associated with practice were treated as stress [11]. However, simply explaining the emotional response experienced during practice as stress makes it difficult to develop a specific intervention strategy. It is more appropriate to focus on the emotional response to change and the relative differences within the educational environment, instead of the broad concept of stress [10]. For nursing students in particular, nurses are very important as part of the clinical practice environment. In Korea, the hierarchy between nurses and nursing students is rigid [12], so nursing students sometimes experience difficulties, and tools that specifically reflect these characteristics are needed. Additionally, since this hierarchical relationship is a unique characteristic of the Confucian cultural sphere, if a tool reflecting this culture is developed, it is expected to be applicable not only to Korea but also to other countries within the Confucian cultural sphere.

Currently, Kim and Shin’s scale [6] is primarily used to measure the transition shock of nursing students in South Korea [10,13,14,15]. Kim et al. [16] modified the Kim and Shin Scale [6] for nursing students. The original tool was designed for nurses and focused on heavy work [16]. Kim and Shin’s tool [6] did not adequately reflect the job differences and responsibility levels between nurses and practical students. While the items were developed by modifying parts similar to those of nurses, the unique characteristics of nursing student practice were not prominently portrayed. 

This study aims to verify the validity and reliability of a newly developed clinical practicum transition shock scale to measure transition shock more accurately by focusing on the characteristics of the practical education of Korean nursing students.

## 2. Methods

### 2.1. Study Design

This methodological study was conducted in two steps: first, a clinical practicum transition shock scale (CPT-shock) for nursing students was developed, and second, its reliability and validity were verified. Phase 1 focused on developing the scale, assessing the content validity, and conducting a pilot study (to test acceptability). Phase 2 was aimed at verifying construct validity, criterion validity, and internal reliability.

### 2.2. Research Procedures

#### 2.2.1. Phase 1: CPT-Shock Tool Development

Composition of Preliminary Questions:

This study was conducted as a follow-up study to a qualitative study by Ko and Kim (2022) [17] that explored the properties of transition shocks in nursing students. Preliminary questions were drafted based on the study by Ko and Kim (2022) on transition shock experience. The sub-factors of the questionnaire tool were also based on the interview content [17] and the transition shock model [18]. The five attributes were role identity confusion, psychological pressure due to an unfamiliar environment, unfriendly interpersonal relationships, disparity between theory and nursing practice, and worries about becoming a clinical nurse. Preliminary questions were placed on each sub-factor. Each of the 12 to 37 preliminary questions was refined into five attributes, and this process was conducted through a discussion process until two nursing professors reached an agreement. The first questionnaire consisted of 104 preliminary questions, and the tool used a 4-point Likert scale (1 = not very much; 4 = very much) to correspond with the existing tool [6].

Verification of Content Validity for Preliminary Questions:

We used the content validity test method as suggested by Lynn (1986) [19]. In order to validate the expert content, an expert group consisting of seven professors and a Doctor of Nursing with more than five years of clinical practice guidance experience was formed. Experts rated each item (very appropriate = 4; not very appropriate = 1) and the content validity index (CVI) was calculated to determine the degree of agreement among experts. In the content validity stage, items with a CVI of 0.8 or higher were selected [19], and items below that were modified or deleted. Each question and the appropriateness of the sub-factor composition were checked, and descriptive answers were prepared for items that had to be corrected or supplemented. Based on the first preliminary tool, the five sub-factors were modified into six sub-factors. Among the initial sub-factors, some of the “psychological pressures from unfamiliar environments” were coded as “changes in daily life patterns” and were separated into two sub-factors. In response to experts’ opinion that “real situation” is a more appropriate code than “unfamiliar environment,” the name of the sub-factor was changed to “pressure felt in the real situation.” Content validity was verified twice, 104 preliminary questions were corrected, duplicates were integrated and deleted, and 59 questions were verified. The CVI value of 59 questions was 0.84~0.96. Finally, the verified items were reviewed for grammatical accuracy by a professor of Korean literature with a Ph.D.

Pilot Study:

The questionnaire contained items that had been verified for content validity, and 24 nursing students were asked to respond. An evaluation was conducted based on the questionnaire response time, readability, and appropriateness of the content, and no words or sentences with ambiguous or inappropriate meanings were identified in the preliminary survey. The responses were completed in approximately 10 min, and the participants expressed no difficulty in understanding the overall content.

#### 2.2.2. Phase 2: CPT-Shock Scale Validation

Participants:

This study was conducted with nursing students who had experience with clinical practice for more than two weeks. Participants were recruited regardless of education year because Kim and Shin’s study [20] did not confirm a difference in transition shock between the 3rd and 4th grades. However, since the aim was to measure the emotional response resulting from the practice, potential participants who had a history of psychiatric treatment for depression or anxiety disorders were excluded. Additionally, students who had only teaching experience (which differs from hospital characteristics) and experience in institutional practice in the local community were excluded. Data from 331 participants were collected for this survey. In factor analysis, the sample size calculation method can largely consider the ‘absolute sample size standard’ and the ‘ratio of number of cases to measured variables (n:p)’. This study was based on an absolute sample size of 200 or more and an n:p ratio of 5:1 [21,22]. Since a five-fold sample size is required to perform exploratory and confirmatory factor analysis on the 59 questions that have been verified for content validity, a total of 259 or more people meets the sample size for statistical testing. Therefore, this study met the standard sample size.

Data Collection and Setting:

Participants were recruited from an unspecified nursing college to recruit participants from as diverse a region as possible. In other words, sampling was performed to represent the characteristics of Korea, not the characteristics of a specific region. Nursing students are at a higher risk of feeling coerced because of their associated hierarchies. Thus, it was thought that collecting data online would counteract this by ensuring autonomy rather than involving every university professor in publicizing and collecting data. Therefore, data were collected by posting promotional materials on Facebook community channels for nursing students. Participants were asked to read the research promotional material and verify the specific research purpose and content by accessing it via the web or an app. If they understood the purpose and content of the study and agreed to it, they were asked to complete the questionnaire after indicating this with a checkbox marked “agree”. Participation in the study took approximately 10–15 min. Due to the nature of online surveys, there is a high possibility of missing responses, so to prevent this, the questionnaire was constructed to terminate the session if all the items were answered.

To reward the participants, those who submitted the questionnaire with more than 90% of the data filled in received a USD 4 online gift card.

Measurements:

To determine the criterion validity of the developed instrument, the transition shock scale was used [6] and adapted to the clinical practicum [23]. Approval was obtained from the tool developer prior to the start of the study. Additionally, to confirm the possibility of generalization, the grade and type of the main practice hospital were collected as general characteristics.

Transition Shock:

Kim and Shin’s (2019) [20] tool was used to measure the shock responses experienced by nursing students during clinical practice, such as doubt, confusion, and loss of support, which were difficult to endure. With a total of 17 items, this tool consists of the “shrinkage of interpersonal relationships,” “conflict between theory and practice,” “confusion of nursing professionalism,” “overwhelming practice work,” “dissonance between personal life and practice,” and “loss of social support”. In a previous study, Cronbach’s α was 0.85 [6], and in this study, it was 0.881.

Adapting to The Clinical Practicum:

Adaptation is negatively correlated with transition shock [10]; it is an indicator of successful transition [24]. Therefore, adaptation to clinical practice was used to verify the criterion validity of the transition shock. In this study, the clinical practice adaptation tool developed by Yi (2007) [23], a single-factor tool consisting of 14 items, was used. At the time of development, Cronbach’s α was 0.86, and in this study, the reliability was 0.655 [23].

### 2.3. Data Analysis

SPSS WIN 23.0 and AMOS 23.0 were used for data analysis in this study. The general characteristics of the participants, which were categorical data, and percentages were calculated using descriptive statistics. Validity, content validity, construct validity, and criterion validity were also verified. CVI was calculated and evaluated using a 4-point scale. Items with a CVI of 0.8 or higher were accepted [19]. Exploratory and confirmatory factor analyses were performed to verify the construct validity. An exploratory factor analysis (EFA) was performed using the varimax and principal component extraction methods. The factor loading, eigenvalue, variance, cumulative variance, Kaiser–Meyer–Olkin (KMO), and Bartlett test results were confirmed. To secure discriminant validity, items with a factor loading of ≤ 0.4 [25] were deleted. Confirmatory factor analysis (CFA) checks X^2^, CMIN/DF, RMSEA, GFI, AGFI, NNFI, and NFI as fit indices for the model and is based on a CMIN/DF of 3 or less [26]. Based on each fit index of 0.8 or higher [26], the model was modified by removing items if the standard value was not reached. The average variance extracted (AVE) was ≥ 0.5 and the construct reliability (CR) was ≥ 0.6 to secure concentrated validity [27]. To verify the criterion validity, the existing transition shock scale, clinical practice adaptation, and Pearson correlation coefficient were obtained. The reliability of the tool was evaluated by calculating Cronbach’s α to confirm its internal consistency.

### 2.4. Ethical Consideration

Approval for this study was obtained from the public IRB (no. P01-202105-21-014) before initiation. Data were collected only from those participants who voluntarily expressed their intention to participate and provided informed consent. The possibility of being excluded or choosing to withdraw from the study was described in detail in the publicity letter. In particular, given that students attending the Research Director’s current school may be considered vulnerable, the recruitment of participants was promoted only through the college bulletin board, and preliminary research or current research promotion was conducted only in the online nursing student community. Additionally, sensitive personal information that could identify an individual, such as name, student number, or date of birth, was not included in the data, and care was taken not to infringe on the autonomy of research participation.

## 3. Results

### 3.1. General Characteristics of Study Participants

The general characteristics of the participants are listed in Table 1. The majority were female (89.4%); 32.3% were in the 3rd grade and 67.7% were in 4th grade. Participants were located in 15 different regions.

### 3.2. Construct Validity of the CPT-Shock Scale for Nursing Students

#### 3.2.1. Exploratory Factor Analysis 

An EFA was performed on the initial 59 items (Table 2). Four exploratory factor analyses found that the KMO test indicated 0.90, Bartlett’s sphericity test indicated χ^2^ = 4919.17 (*p* < 0.001), and the correlation coefficient matrix was found to be suitable for the absolute value of the maximum factor loading, which varies from 0.30 to 0.50 [25], but in this study, items with a value ≥ 0.40 were selected.

The factor analysis results were classified into six factors with a cumulative explanatory power of 52.72%. Cumulative explanatory variance indicates the significance of meaning and contribution to the total factor variable, and since social science studies typically recommend 50% to 60% [25], the results of this study met the minimum standard. The factor loadings for each factor range from 0.41 to 0.77 and the variance explained by the factors was 13.65% for Factor 1, 9.73% for Factor 2, 8.96% for Factor 3, and 8.96% for Factor 4. Factor 5 was 7.03% and Factor 6 was 5.09%. Factor 1 was “unfriendly interpersonal relationships,” Factor 2 was “changes in daily life patterns,” Factor 3 was “worries about becoming a clinical nurse,” Factor 4 was “pressure felt in real situations,” Factor 5 was “disparity between theory and nursing practice,” and Factor 6 was named “role identity confusion”.

#### 3.2.2. Confirmatory Factor Analysis

A CFA was performed on the clinical practice transition shock of nursing students extracted through an EFA (Table 3, Figure 1). Items with a factor loading of > 0.50 and significance (C.R. > 1.96, *p* < 0.05) [28] were selected. Of the 38 derived items, 19 were derived after deleting items that exceeded the standard value of 50 [29] based on a modification index (MI). The fit of the model was CMIN = 238.471, CMIN/DF = 1.741, *p* < 0.001, GFI = 0.930, AGFI = 0.902, TLI = 0.923, CFI = 0.938, and RMR = 0.035.

#### 3.2.3. CPT-Shock Convergence and Discriminant Validity for Nursing Students

The convergence and discriminant validity are presented in Table 3. Concentration or convergence validity was confirmed by satisfying the criteria (AVE > 0.50) with AVE 0.50–0.67, and CR 0.75–0.83 (CR > 0.70). Discriminant validity is used to verify whether the independence and low correlation between the sub-factors of the tool are shown and whether the AVE of each latent variable is greater than the square of the correlation coefficient between the latent variables. As a result, in the correlation coefficient range (0.22 to 0.63), the value of the square of the correlation coefficient (0.048 to 0.39) was lower than the AVE value (0.50 to 0.67). This meant that the constructs differed by maintaining a low correlation and independence among the six sub-factors of the tool, and discriminant validity was established.

### 3.3. Criterion Validity of the CPT-Shock Scale for Nursing Students

To verify the criterion validity, the correlation between the existing transition shock scale and adaptation to clinical practice was verified (Table 4). Validity was positively correlated with existing tools (r = 0.779, *p* < 0.001) and negatively correlated with adaptation to clinical practice (r = −0.505, *p* < 0.001).

### 3.4. Descriptive Statistics and Reliability of the CPT-Shock Scale for Nursing Students

The descriptive statistics of the final confirmed items are presented in Table 5. The average of the items was 2.80 out of 4 points. Reliability was confirmed by calculating the tool’s Cronbach’s α (α = 0.85) (Table 5), which was found to be at a good level.

## 4. Discussion

Based on Duchscher’s transition shock model [18], this study developed questions by interviewing nursing students with clinical practice experience and verifying the content validity, construct validity, and reliability of the CPT-shock scale for nursing students. 

This study is meaningful in that it has developed a tool to measure nursing students’ unique experiences. These students changed their practice sites for each practice and had different role characteristics than nurses. Moreover, given that the items reflect the real student experience of clinical nursing education, it is believed that the tool can measure nursing students’ transition shock more accurately than existing ones. The final tool developed in this study comprised six sub-factors and 19 items. The six sub-factors were “unfriendly interpersonal relationships”, “changes in daily life patterns”, “worries about becoming a clinical nurse,” “pressure felt in the real situation,” “disparity between theory and nursing practice”, and “confusion about role identity”. 

First, “unfriendly interpersonal relationships” in Factor 1 refers to the formation of unfamiliar and uncomfortable relationships with people in the work environment. Four questions were developed, including questions regarding levels of interest in practical training and the ease of receiving help. This factor corresponds to the attributes “look for familiar protective nurturing”, “require positive reinforcement”, and “lack of support” among the emotional domains in the transition shock model [18]. Students have the opportunity to learn nursing skills and procedures from nurses through hands-on practice and the desire to learn to communicate effectively with patients and their families [30]. However, nurses must supervise students while performing their own tasks; their role as student educators is not mandatory and they may show negative attitudes due to a lack of incentives [12]. Specifically, it is difficult to establish a positive and effective relationship between students who must follow nurses constantly during the clinical practicum and nurses who are burdened with student education [12,31]. In other words, because nurses lack the environmental conditions or motivation to educate nursing students, their attitude toward nursing students appears to be neither warm nor supportive. Unlike educational institutions, the difficulty of building positive relationships with nurse educators can be a factor in transition shock.

Factor 2, “changes in daily life patterns”, refers to experiencing changes throughout one’s personal life, including academics, while receiving clinical practice education. Three questions were developed, including those on the physical burden and concentration. This area corresponded to “energy consumed to conceal feelings and transition responses” and “change in social habits and routines” among the physical domains in the transition shock model [18]. Nursing students responded that it was more emotionally and physically difficult than the theory class because they had to practice with the mindset that they should not make mistakes [12,32]. Additionally, although nursing students perform simple tasks during the practice period, they must adapt to the shift work schedule, and lack of rest time adds to their physical burden [12]. The physical and psychological burden during the practice period appears to affect daily life; therefore, it is thought to reflect nursing students’ reality well. Because the tasks given during the practice period are also a burden to students [32], it is thought that the original daily pattern cannot be maintained during the training period; therefore, it can be considered as an appropriate transition shock factor. 

Factor 3, “Worries about becoming a clinical nurse”, is a negative after-effect of clinical practice that makes students reconsider their decision to become professional nurses. Three items were developed, including those testing self-esteem and concerns about future careers. This factor corresponds to the “oppressive hierarchical work structure” and “insufficient exposure to role models” among the socio-developmental domains of the transition shock model [18]. In the process of adapting to the clinical setting, nursing students’ practice corresponds to the stage just before entry into the profession. Thus, it is considered to contain meaningful information about the transition shock stage. This is because the experiential value of clinical practice motivates students to pursue careers as clinical nurses [32]. Nursing students sometimes indicate that it is difficult because they remember their experiences during practice, even after the practice has ended, which leads to a buildup of emotions. In this study, students expressed a desire for the practice period, as opposed to a desire to learn [32]. Additionally, if they experience the absence of a role model indirectly or the formation of a hierarchical relationship, nursing students feel burdened by their future career choices as nurses [12]. This is considered an appropriate factor to measure in the transition phase, as negative perceptions can first be established through practice before entering the clinical field.

Factor 4, “Pressure felt in real situations”, includes the burden, fear, and anxiety the students feel when they are in a real situation and not dealing with a model or standard patient, as opposed to what they experience in school. Three questions were developed concerning tension, worry about mistakes, and fear. This factor corresponds to “fear of failure or incompetence” and “extreme sensitivity” among the emotional domains of the transition shock model [18]. In clinical practice, most nursing students experience emotional agitation and anxiety when they encounter patients [32]. This stems from the fear of making a mistake or harming patients through their actions [12]. This is a common reaction when students start practicing and is believed to be inevitable because the school does not train nursing students with real patients. 

Factor 5, “Disparity between theory and nursing practice”, refers to the confusion and doubt experienced due to inconsistencies between the theoretical knowledge acquired at the college and the nursing practice performed by actual nurses. Four items were developed, including different protocols based on hospitals and nurses and differences from the theory learned at college. This factor corresponded to “limited tacit/practical knowledge” and “limited practice aspect, pattern recognition” among the intellectual domains of the transition shock model [18]. One of the main advantages of clinical practice is that it allows the acquisition of a wide range of medical knowledge and an understanding of nursing through interactions with real patients [33]. However, students are sometimes confused because the theoretical knowledge they have learned at school differs from the practice [12]. Students indicate that because of individual differences among nurses who teach, every time the instructing nurse changes, what they learn changes [31]. Because students lack clinical experience, they find it difficult to accept nursing procedures that may vary depending on their clinical situation. Therefore, there can be confusion in the process of expanding existing nursing knowledge; this attribute has also been included in the transition shock factor.

Factor 6, “role identity confusion”, refers to the ambiguity of nursing students’ scope of work. Two questions were developed to assess whether the respondents recognized the role of a nursing assistant and treated it as such. This factor corresponds to “role uncertainty and familiarity” and “inadequate and insufficient guidance/assistance” among the socio-developmental domains of the transition shock model [18]. Nursing students spend most of their time standing up and observing their patients during practice [33]. Additionally, students recognize that their nursing practice is limited because they do not have a nurse’s license, but they want to practice nursing under the supervision of nurses [33]. New nurses recognize that they will continue to associate with other medical personnel or people from related departments in the future, while students only stay for a short period of time (generally a week) [12]. Thus, there is a lack of imprint as future colleagues need to be taught and guided. Students are expected to play the role of medical personnel for guardians or patients in clinical practice; however, they may not be able to cope because of a lack of knowledge or practical experience [32]. Thus, nursing students are limited to mere work assistant roles and have doubts about their roles. Because some hospitals lack guidelines for student education [12], the extent of student roles is not specified, and students may experience role confusion.

Questions for this tool have been developed to measure nursing students’ career skepticism, the placement of trainees (role identity), and interpersonal relationships between students and employees, which are difficult to measure using existing tools. As nursing students are not in the clinical stage, unlike new nurses, questions of career skepticism were constructed considering cases where they did not enter the clinical setting from the outset. In particular, since most nursing students mainly observe during practice, they are far from dealing with a heavy workload or performing diverse tasks. Unlike existing tools that include these items, items expressing the fear or tension of meeting a real patient are more suitable for measuring transition shock in nursing students. Additionally, in the newly developed tool, the ambiguous role of a nursing student corresponds to reality, as nursing students remain in practice temporarily. The sense of disparity between theory and nursing behavior on the ground was similar to that of the existing tools. 

Since this study targeted nursing students in Korea, it is believed that there will be limitations in applying this scale internationally as it did not consider various factors such as academic/clinical practice system, and cultural background. However, it is believed that the CPT-shock scale can be applied to Asian countries that have similar clinical practice environments or cultural environments to Korea and is expected to contribute to evaluating the clinical practice environment in Asia in the future.

## 5. Conclusions

This study aimed to develop a tool to quantitatively measure the transition shock experienced by nursing students in clinical practice and to verify its validity and reliability. The tool was developed as a 4-point Likert scale questionnaire with six sub-factors and 19 items. The higher the total score, the higher the level of transition shock. This tool can be used to evaluate the transition shock level of nursing students during clinical practice and should be helpful in predicting the ability to adapt to clinical practice in the future. Since the transition shock of nursing students can affect their future career path and clinical adaptation, it is suggested that it be used for clinical practice environment evaluation or the development of clinical practice education guidelines to advance clinical practice education.

## Figures and Tables

**Figure 1 healthcare-11-02789-f001:**
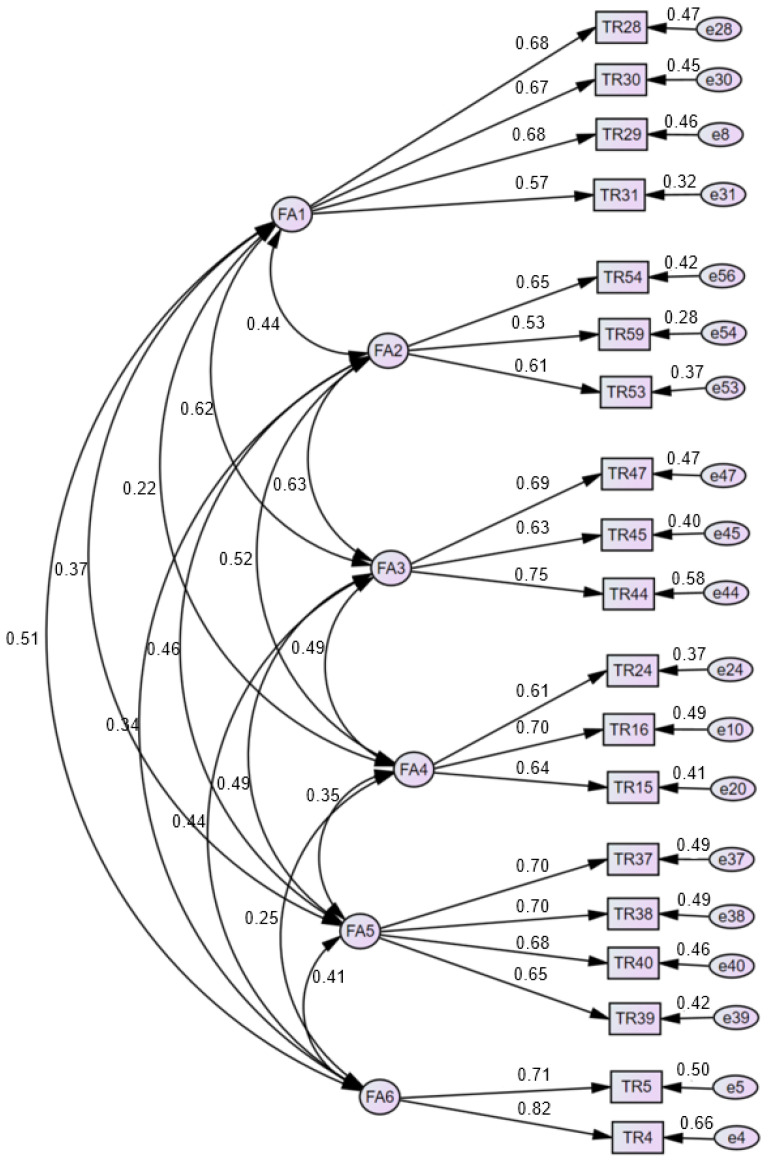
Confirmatory factor analysis findings and final items.

**Table 1 healthcare-11-02789-t001:** General characteristics of participants (N = 331).

Characteristics	Categories	N (%)
Sex	Male	35 (10.6)
	Female	296 (89.4)
Education year	Third-year	107 (32.3)
	Fourth-year	224 (67.7)
Type of training hospital	Higher general hospital	167 (50.5)
	General hospital	162 (48.9)
	Others	2 (0.6)
Location of the college	Gyeongsangbuk-do	51 (15.4)
	Busan	35 (10.6)
	Gyeongsangnam-do	33 (10.0)
	Chungcheonnam-do	30 (9.1)
	Deagu	29 (8.8)
	Gyeonggi-do	27 (8.2)
	Jeollabuk-do	25 (7.6)
	Gwangju	19 (5.7)
	Chungcheonbuk-do	17 (5.1)
	Daejeon	14 (4.2)
	Seoul	13 (3.9)
	Gangwon-do	11 (3.3)
	Jeollanam-do	10 (3.0)
	Ulsan	9 (2.7)
	Incheon	3 (0.9)
	No reported	5 (1.5)

**Table 2 healthcare-11-02789-t002:** Exploratory factor analysis (N = 331).

Factors	Items	Factors					
		1	2	3	4	5	6
Factor 1	31	0.678					
	30	0.66					
	29	0.643					
	28	0.638					
	33	0.628					
	27	0.609					
	35	0.596					
	26	0.591					
	36	0.589					
	32	0.576					
	34	0.549					
	8	0.513					
Factor 2	58		0.687				
	52		0.66				
	53		0.642				
	57		0.641				
	56		0.6				
	54		0.593				
	59		0.557				
	50		0.501				
Factor 3	47			0.769			
	49			0.697			
	46			0.638			
	45			0.574			
	44			0.553			
	48			0.481			
Factor 4	24				0.716		
	16				0.671		
	15				0.666		
	20				0.596		
	21				0.488		
	25				0.412		
Factor 5	37					0.769	
	38					0.75	
	40					0.73	
	39					0.646	
Factor 6	5						0.646
	4						0.577
Eigenvalue		9.96	3.04	1.95	1.88	1.81	1.39
Total Variance Explained (%)		13.65	9.72	8.96	8.25	7.03	5.09
Cumulative Variance (%)		13.65	23.38	32.34	40.59	47.63	52.72

Factor 1 = “unfriendly interpersonal relationships”; Factor 2 = “changes in daily life patterns”; Factor 3 = “worries about becoming a clinical nurse”; Factor 4 = “pressure felt in real situations”; Factor 5 = “disparity between theory and nursing practice”; Factor 6 = “role identity confusion”.

**Table 3 healthcare-11-02789-t003:** Confirmatory factor analysis findings and final items (N = 331).

Factors	Items	β	S.E.	C.R.	Factor						AVE	CR
					1	2	3	4	5	6		
					r							
Factor 1	28	0.685				0.44	0.62	0.22	0.37	0.51	0.54	0.83
	30	0.674	0.115	9.666								
	29	0.68	0.104	9.722								
	31	0.569	0.107	8.493								
Factor 2	53	0.607					0.63	0.53	0.46	0.34	0.50	0.75
	59	0.528	0.156	6.732								
	54	0.646	0.159	7.451								
Factor 3	47	0.687						0.49	0.49	0.44	0.50	0.75
	45	0.63	0.086	9.332								
	44	0.745	0.098	10.408								
Factor 4	24	0.61							0.49	0.26	0.50	0.75
	16	0.699	0.117	7.904								
	15	0.639	0.109	7.743								
Factor 5	37	0.697								0.41	0.50	0.80
	38	0.697	0.108	10.151								
	40	0.678	0.107	9.964								
	39	0.648	0.096	9.633								
Factor 6	5	0.708									0.67	0.80
	4	0.815	0.144	7.813								

Model Fitness: χ^2^ = 238.47 (*p* < 0.001), χ^2^/df = 1.741, GFI = 0.930, AGFI = 0.902, NFI = 0.869, TLI = 0.923, CFI = 0.938, RMR = 0.035, RMSEA = 0.047; AGF = adjusted goodness-of-fit index; AVE = average variance extracted; CFI = comparative fit index; CR = construct reliability; GFI = goodness-of-fit index; NFI = normed fit index; RMSEA = root mean square error of approximation; TLI = Tucker–Lewis index. Factor 1 = “unfriendly interpersonal relationships”; Factor 2 = “changes in daily life patterns”; Factor 3 = “worries about becoming a clinical nurse”; Factor 4 = “Pressure felt in real situations”; Factor 5 = “Disparity between theory and nursing practice”; Factor 6 = “role identity confusion”.

**Table 4 healthcare-11-02789-t004:** Criterion validity of the CPT-shock scale.

Transition Shock	Clinical Practicum Adaptation; r (*p*)
0.779 (*p* < 0.001)	−0.505 (*p* < 0.001)

**Table 5 healthcare-11-02789-t005:** Descriptive statistics, reliability of items (N = 331).

Items		Actual Range	Mean ±SD	Cronbach’s α if Item Deleted	Cronbach’s α
28	I felt that the clinical practice field leader (head nurse) was not particularly interested in the students in practice.	1.0–4.0	2.71 ± 0.86	0.84	0.74
30	Even if I did not know or had any questions, I could not easily ask the nurse.	1.0–4.0	2.60 ± 0.89	0.84	
29	I felt out of place in the training ward and was wandering around.	1.0–4.0	2.58 ± 0.80	0.84	
31	I felt there was hardly anyone to help me when I was in trouble in the ward where I was practicing.	1.0–4.0	2.24 ± 0.79	0.845	
53	The clinical practicum period was physically harder for me than the theory class.	1.0–4.0	3.45 ± 0.79	0.84	0.61
59	It was difficult to concentrate anywhere because I had to deal with both the ward training and the given assignments.	1.0–4.0	2.75 ± 0.88	0.84	
54	I have experienced going to work earlier or leaving work late because of the burden of practice.	1.0–4.0	2.85 ± 0.96	0.84	
47	After the practice, I thought that becoming a nurse was not the right path for me.	1.0–4.0	2.25 ± 0.89	0.83	0.73
45	After the practice, I thought, “Can I work with pride as a nurse in the future?”	1.0–4.0	2.84 ± 0.77	0.84	
44	I felt less self-esteem rather than feeling that I had grown after the practice.	1.0–4.0	2.18 ± 0.84	0.838	
24	I was afraid of being reprimanded for my unskilled skills during practice.	1.0–4.0	2.88 ± 0.81	0.85	0.67
16	I was worried that my mistake during the practice might harm the patient.	1.0–4.0	3.33 ± 0.65	0.85	
15	Being a real patient, I get nervous even when doing trivial things.	1.0–4.0	3.28 ± 0.65	0.84	
37	I was confused because there was a difference (modification or omission of procedures) between the theory I learned and the nursing skills observed in the ward.	1.0–4.0	2.84 ± 0.68	0.84	0.77
38	It was difficult for me to accept that the techniques were slightly different for each nurse, ward, and hospital.	1.0–4.0	2.73 ± 0.74	0.84	
40	I wondered why the theory I had learned and the practice I experienced in practice were different.	1.0–4.0	2.70 ± 0.74	0.84	
39	I was confused because I didn’t know which one to consider first when the theoretical guidelines I knew and the clinical situation conflicted.	1.0–4.0	2.91 ± 0.67	0.84	
5	I felt that nursing students in the ward were mainly perceived as nursing assistants (e.g., assistants, transfer agents).	1.0–4.0	3.03 ± 0.84	0.84	0.73
4	I spent most of my practice time assisting nurses’ chores (e.g., patient transfer, specimen/drug transport).	1.0–4.0	3.13 ± 0.82	0.84	
Total (19 items)		1.0–4.0	2.80 ± 0.41		0.85

M = mean; SD = standard deviation; Factor 1 = “unfriendly interpersonal relationships”; Factor 2 = “changes in daily life patterns”; Factor 3 = “worries about becoming a clinical nurse”; Factor 4 = “Pressure felt in real situations”; Factor 5 = “disparity between theory and nursing practice”; Factor 6 = “role identity confusion”.

## Data Availability

The data presented in this study are available on request from the corresponding author.

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
