# Peer review of "Development and Validation of a Clinical Practicum Transition Shock Scale (CPT-Shock) for Korean Nursing Students"

_healthcare, 2023, doi:10.3390/healthcare11202789_

Round 1

Author Response

Thank you for your very helpful comments.
We have revised the manuscript according to your comments.

Reviewer 2 Report

Thank you for the opportunity to review the article. The study aimed to develop a tool for nursing students' adaptation to practical classes in Korean clinics. Nursing students experience many changes during the transition from theoretical to practical training. "Transition shock" is an emotional reaction that can occur during the transition process and includes confusion, doubt and loss of support. The study targeted 6 key factors to reduce adverse reactions associated with practical classes in clinics and direct patient contact.
My suggestions to the authors:

  • The assessment of attitudes of active nurses toward nursing students described in factor 1 during practical classes;

  • Limitations - A limitation of this study was subjectivity of answers too

Author Response

(The authors gave the same response as above.)

Reviewer 3 Report

Thank you for the opportunity to explore such exciting research.

The methodology is described in sufficient detail, the article's purpose is clearly stated, and the results are also quite logical

The authors also emphasized the theoretical and practical significance of the research, so I literally have a couple of suggestions to improve the presentation of the research:

1. The concept of clinical practice transition shock should be defined and its attributes should be confirmed. There is no definition of the concept in this study. What is the reason for using Ko and Kim's research as a base? Why did you choose it ? And the study [17] is a model for nurses. The process of in-depth analysis of previous studies dealing with the clinical practice transition shocks of nursing students seems to have been omitted. I think the research process needs to be well summarized.

2. Line 92: Explain in detail what the researchers agreed to. And please give the CVI value

3. Line 97 : 6 attributes are not specific. Is 1 extra from 5? Is there any change from 5? Is it the same as the factor suggested in Line 207~211?

4. .No data analysis references were provided.

5. Line 215-218: What was the researcher's subjective assessment?

6. Please enter the factor name at the bottom of Tables 2, 3, and 5.

I hope that the author will do his best to make the necessary corrections to the article for really interesting research to be published.

Author Response

Thank you for your helpful comment. As your recomment, we revised manuscript. 
